# SIReN-VAE: Leveraging Flows and Amortized Inference for Bayesian Networks

**Jacobie Mouton**
Computer Science Division
Stellenbosch University
South Africa

**Steve Kroon**[*]
Computer Science Division
Stellenbosch University AND
National Institute for Theoretical and Computational Sciences
South Africa

## ABSTRACT

Initial work on variational autoencoders assumed independent latent variables with simple distributions. Subsequent work has explored incorporating more complex distributions and dependency structures: including normalizing flows in the encoder network allows latent variables to entangle non-linearly, creating a richer class of distributions for the approximate posterior, and stacking layers of latent variables allows more complex priors to be specified for the generative model. This work explores incorporating *arbitrary* dependency structures, as specified by Bayesian networks, into VAEs. This is achieved by extending both the prior and inference network with graphical residual flows—residual flows that encode conditional independence by masking the weight matrices of the flow's residual blocks. We compare our model's performance on several synthetic datasets and show its potential in data-sparse settings.

## 1 INTRODUCTION

Variational autoencoders (VAEs) (Kingma & Welling, 2014; Rezende et al., 2014) provide a powerful framework for constructing deep latent variable models. By positing and fitting a generic model of the data generating process, they allow one to generate new samples and to reason probabilistically about the data and its underlying representation. Despite the success of VAEs, an early shortcoming identified is that they typically make use of overly simple latent variable distributions, e.g. using fully-factorized Gaussian distributions as both the prior and approximate posterior over the latent variables. Subsequent work has explored incorporating more complex latent variable distributions and have shown that this results in improved performance: normalizing flows can be included as part of the VAE's encoder network (Kingma et al., 2016), entangling the latent variables non-linearly to obtain a richer class of approximate posterior distributions. The prior distribution can also be made more complex, for example by stacking layers of latent variables to create a hierarchical structure (Sønderby et al., 2016). This increases the flexibility of the true posterior, leading to improved empirical results (Kingma et al., 2016).

Traditional VAEs, as well as those with flow-enriched inference networks, do not allow one to directly control the dependence structure encoded by the model. Stacked latents can encode hierarchical dependencies, but limit us to simple conditional distributions. This work proposes an approach to incorporating rich conditional distributions for *arbitrary* dependency structures—specified by Bayesian networks (BNs)—into VAE models. The method extends both the prior and inference network with *graphical* residual flows, which encode the dependence structure by masking the weight matrices of the flow's residual blocks to enforce sparsity. The resulting model can thus learn mappings between a simple distribution and more complex distributions that conforms to this dependence structure. We evaluate the performance of our approach by comparing the effects of encoding different dependencies on several synthetic datasets. We find that encoding the dependency information from the true BN associated with the data yields better results than other approaches in data-sparse settings; however, no clear advantage is observed when large datasets are available.

---

[*]Correspondence to `kroon@sun.ac.za`.

## 2 VAEs with Structured Invertible Residual Networks

If we have prior knowledge about the data generating process, it seems likely to be beneficial to incorporate this knowledge in a VAE. In this work, we assume access to a Bayesian network (BN) specifying the dependency structure over $D$ observed and $K$ latent variables. Our goal is to suitably incorporate this dependency information into the VAE's encoder and decoder networks. Using $\theta$ for the decoder network's parameters, this means that its likelihood $p_\theta(\mathbf{x}|\mathbf{z})$ and prior $p_\theta(\mathbf{z})$ should factorize according to the BN's conditional independencies. Approximating the posterior distribution $p(\mathbf{z}|\mathbf{x})$ while taking the knowledge from the BN into account requires suitably inverting the BN such that one obtains edges from $\mathbf{x}$ to $\mathbf{z}$. Webb et al. (2018) showed the importance of encoding the generative model's true inverted structure in the VAE's encoder and provide an algorithm for obtaining a suitable minimal faithful inverse of a BN.

### 2.1 SIReN-VAE

We use graphical residual flows (GRFs; see section 2.2) to incorporate structure in a VAE's latent space, yielding the structured invertible residual network (SIReN) VAE. For an observed sample $\mathbf{x}$, the encoder network, with parameters $\phi$, is defined as a flow conditioned on $\mathbf{x}$:

$$\mathbf{z} = \text{GRF}_g(\boldsymbol{\epsilon}; \mathbf{x}, \phi) \quad \text{where} \quad \boldsymbol{\epsilon} \sim p_0 \tag{1}$$

$$\log q_\phi(\mathbf{z}|\mathbf{x}) = \log p_0(\boldsymbol{\epsilon}) - \log \left| \det(J_{\text{GRF}_g}(\boldsymbol{\epsilon})) \right| \tag{2}$$

The subscript $g$ here denotes that this is a generative flow, $\det(J_F(\cdot))$ denotes the Jacobian determinant of a flow transformation $F$, and we set $p_0$ to $\mathcal{N}(\mathbf{0}, I)$. For a sample $\mathbf{z}$ from the encoder network, the decoder uses a normalizing flow for the prior density, and a fully-factored Gaussian likelihood, with parameters output by a network denoted by DecoderNN:

$$\log p_\theta(\mathbf{z}) = \log p_0(\text{GRF}_n(\mathbf{z}; \theta)) + \log \left| \det(J_{\text{GRF}_n}(\mathbf{z})) \right| \tag{3}$$

$$\boldsymbol{\mu}, \log \boldsymbol{\sigma} = \text{DecoderNN}(\mathbf{z}; \theta) \tag{4}$$

$$p_\theta(\mathbf{x}_i|\mathbf{z}) = \mathcal{N}(\mathbf{x}_i; \mu_i, \sigma_i), \quad i = 1, \ldots, D \tag{5}$$

The subscript $n$ above denotes a normalizing flow. Figure 1 illustrates the full VAE. Note that $\text{GRF}_n$ must be inverted to generate samples from this VAE, making sample generation slower than with regular VAEs.

### 2.2 Graphical Residual Flows

Graphical flows (Wehenkel & Louppe, 2021; Weilbach et al., 2020) add further structure to normalizing flows (NFs) (Tabak & Turner, 2013; Rezende & Mohamed, 2015) by encoding non-trivial variable dependencies through sparsity of the neural networks' weight matrices. While the graphical flows of Wehenkel & Louppe (2021) or Weilbach et al. (2020) could also be used, we instead consider applying similar ideas to residual flows (Chen et al., 2019). We choose the resulting GRFs over other graphical flows due to their faster and more stable inversion behaviour, as discussed in Behrmann et al. (2021); Mouton & Kroon (2022). Consider a residual network $F(\boldsymbol{y}) = (f_T \circ \ldots \circ f_1)(\boldsymbol{y})$, composed of blocks $\boldsymbol{y}^{(t)} := f_t(\boldsymbol{y}^{(t-1)}) = \boldsymbol{y}^{(t-1)} + g_t(\boldsymbol{y}^{(t-1)})$. $F$ is an NF if all of the $f_t$ are invertible. A sufficient condition for invertibility of $f_t$ is $\text{Lip}(g_t) < 1$, where $\text{Lip}(\cdot)$ denotes the Lipschitz constant of a transformation. Behrmann et al. (2019) construct a *residual flow* by applying spectral normalization to the residual network's weight matrices such that the bound $\text{Lip}(g_t) < 1$ holds for all layers. The graphical structure of a BN can be incorporated into a residual flow by suitably masking the weight matrices of each residual block before applying spectral normalization.

**Normalizing GRF** Given a BN graph, $\mathcal{G}$, the update to $\mathbf{z}^{(t-1)}$ in block $f_t$ of $\text{GRF}_n$ in the decoder is defined as follows for a residual block with a single hidden layer (it is straightforward to extend this to residual blocks with more hidden layers):

$$\mathbf{z}^{(t)} := \mathbf{z}^{(t-1)} + (W_{d,2} \odot M_{d,2}) \cdot h((W_{d,1} \odot M_{d,1}) \cdot \mathbf{z}^{(t-1)} + b_{d,1}) + b_{d,2} \ . \tag{6}$$

Here, $h(\cdot)$ is a nonlinearity with $\text{Lip}(h) \leq 1$, $\odot$ denotes element-wise multiplication, $d$ denotes that this operation is part of the decoder, and the $M_{d,i}$ are binary masks ensuring that component $j$ of the

Table 1: Negative Log-likelihood (NLL) and reconstruction error (RE) achieved by each model when trained on different sized training sets. Each entry corresponds to the average performance on 100 test samples over 5 independent runs with standard deviation given in the subscript. Lower is better. Bold indicates the best result in each group. The number of observed (D) and latent (K) variables, as well as the number of edges (E) in the datasets' associated BN are also provided.

| | D | K | E | Model | $2 \times |\mathcal{G}|$ training samples | | $100 \times |\mathcal{G}|$ training samples | |
| --- | --- | --- | --- | --- | --- | --- | --- | --- |
| | | | | | NLL | RE | NLL | RE |
| EColi70 | 29 | 15 | 59 | VAE | $42.99_{\pm 1.50}$ | $6.21_{\pm .32}$ | $36.95_{\pm .01}$ | $4.68_{\pm .01}$ |
| | | | | SIReN-VAE$_{\text{ind}}$ | $43.77_{\pm 0.16}$ | $6.32_{\pm .08}$ | $35.84_{\pm .23}$ | $4.07_{\pm .10}$ |
| | | | | SIReN-VAE$_{\text{FC}}$ | $42.91_{\pm 1.04}$ | $6.03_{\pm .29}$ | $\mathbf{35.77_{\pm .23}}$ | $4.05_{\pm .09}$ |
| | | | | SIReN-VAE$_{\text{true}}$ | $\mathbf{38.98_{\pm 0.81}}$ | $\mathbf{4.95_{\pm 0.22}}$ | $36.22_{\pm .19}$ | $\mathbf{3.88_{\pm .10}}$ |
| Arth150 | 67 | 40 | 150 | VAE | $74.13_{\pm 3.27}$ | $6.13_{\pm .70}$ | $38.92_{\pm .06}$ | $4.50_{\pm .02}$ |
| | | | | SIReN-VAE$_{\text{ind}}$ | $42.60_{\pm 0.38}$ | $4.85_{\pm .02}$ | $\mathbf{38.84_{\pm .11}}$ | $4.45_{\pm .02}$ |
| | | | | SIReN-VAE$_{\text{FC}}$ | $42.15_{\pm 0.00}$ | $4.82_{\pm .00}$ | $39.06_{\pm .07}$ | $4.49_{\pm .01}$ |
| | | | | SIReN-VAE$_{\text{true}}$ | $\mathbf{42.06_{\pm 0.40}}$ | $\mathbf{4.80_{\pm .01}}$ | $38.86_{\pm .22}$ | $\mathbf{4.42_{\pm .04}}$ |
| Magic-Irri | 5 | 59 | 102 | VAE | $15.70_{\pm 1.89}$ | $11.95_{\pm 2.58}$ | $10.18_{\pm .21}$ | $\mathbf{7.30_{\pm .60}}$ |
| | | | | SIReN-VAE$_{\text{ind}}$ | $14.66_{\pm 1.08}$ | $16.66_{\pm 1.40}$ | $10.03_{\pm .00}$ | $9.12_{\pm .01}$ |
| | | | | SIReN-VAE$_{\text{FC}}$ | $12.41_{\pm 3.34}$ | $12.05_{\pm 3.99}$ | $10.03_{\pm .00}$ | $9.13_{\pm .01}$ |
| | | | | SIReN-VAE$_{\text{true}}$ | $\mathbf{10.83_{\pm 0.64}}$ | $\mathbf{11.13_{\pm 1.95}}$ | $\mathbf{10.02_{\pm .01}}$ | $8.91_{\pm .29}$ |

residual block's output is only a function of the input corresponding to $\{z_j\} \cup Pa_{\mathcal{G}}(z_j)$. By composing a number of such blocks, each variable ultimately receives information from its ancestors in the BN via its parents. This is similar to the way information propagates between nodes in a message passing algorithm. The masks above are constructed according to a variant of MADE (Germain et al., 2015) for arbitrary graphical structures. Similar masks are applied to DecoderNN such that $\mu_i$ and $\log \sigma_i$ are only a function of those $z_j \in Pa_{\mathcal{G}}(x_i)$. The change in density incurred by the *normalizing* flow (that maps samples from the data distribution to samples from a base distribution $p_0$) is tracked via the change-of-variable formula (3). Since we are enforcing a DAG dependency structure between the variables, there is a permutation of the components of $\mathbf{z}$ for which the corresponding permuted Jacobian is lower triangular. We can thus compute $\det(J_{\text{GRF}_n}(\mathbf{z}))$ *exactly* as the product of its diagonal entries, since the determinant is invariant under such permutations. The inverse of this flow does not have an analytical form (Behrmann et al., 2019). Instead, each block is inverted numerically using the Newton-like fixed-point method proposed by Song et al. (2019).

**Generative GRF** For the encoder, we determine the structure to embed in the flow by inverting the BN graph using the faithful inversion algorithm of Webb et al. (2018). This leads to a *generative* flow (that provides a mapping from the base to the approximate posterior distribution) where the latents are conditioned on the observations:

$$\mathbf{z}^{(t)} := \mathbf{z}^{(t-1)} + (W_{e,2} \odot M_{e,2}) \cdot h((W_{e,1} \odot M_{e,1}) \cdot \mathbf{y}^{(t-1)} + b_{e,1}) + b_{e,2},$$

where the subscript $e$ denotes the encoder and $\mathbf{y}^{(t-1)} = \mathbf{z}^{(t-1)} \oplus \mathbf{x}$ where $\oplus$ denotes concatenation.

## 3 EXPERIMENTS

We evaluate the performance of SIReN-VAE on datasets generated from three fully-specified BNs, obtained from the BN repository of Scutari (2022) . All leaf nodes were considered observed, and the rest taken to be latent. To better compare the effect of the encoded dependency structure, we train three different SIReN-VAE models. The first encodes a BN with conditionally independent latent variables in the decoder (denoted by SIReN-VAE$_{\text{ind}}$), much like a vanilla VAE (though the prior distributions would be non-Gaussian, unlike a vanilla VAE). The second model, SIReN-VAE$_{\text{FC}}$, encodes a fully-connected structure between the latent variables of the generative model using the same topological ordering as the true BN. Each observed variable is conditioned on all latent variables in both these models. The final model encodes the true dependencies as specified by each dataset's accompanying BN and is denoted by SIReN-VAE$_{\text{true}}$. Both SIReN-VAE$_{\text{ind}}$ and SIReN-VAE$_{\text{FC}}$ use

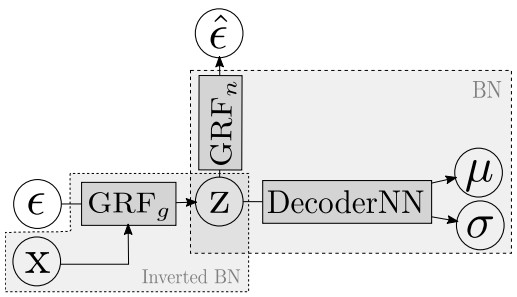
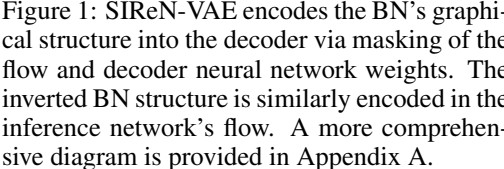

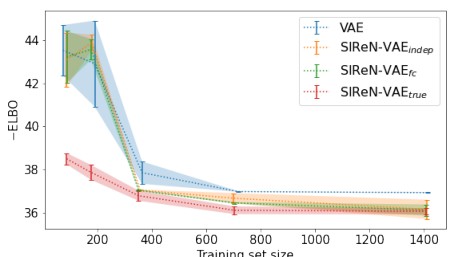

Figure 1: SIReN-VAE encodes the BN's graphical structure into the decoder via masking of the flow and decoder neural network weights. The inverted BN structure is similarly encoded in the inference network's flow. A more comprehensive diagram is provided in Appendix A.

Figure 2: Negative ELBO (lower is better) vs training set size for the EColi70 dataset. Error bars show one standard deviation from the mean over 5 independent runs.

the same latent dimension as the true BN. Note that $GRF_g$ should be deep enough to allow information from $\mathbf{x}$ to reach all latents via their parents. We used four residual blocks for all GRFs in our networks, since this was sufficient; each residual block had a single hidden layer. We also compare the results to those of a vanilla VAE (with fully-factorized Gaussian prior and approximate posterior of the same latent dimension). The encoder and decoder neural networks of this VAE had similar architectures to the DecoderNN used in the SIReN-VAE models. All models were trained using Adam with different initial learning rates (either $10^{-1}$, $10^{-2}$ or $10^{-3}$) based on which resulted in the lowest loss. The learning rate was decreased by a factor of 10 each time no improvement in the loss was observed for 10 consecutive epochs, until a minimum rate of $10^{-6}$ was reached.

We first trained all models on a training set consisting of $100 \times |\mathcal{G}|$ data points where $|\mathcal{G}| = D + K$ is the number of vertices in the BN. We compare the models based on their log-likelihood $p_\theta(\mathbf{x})$ and reconstruction error on 100 test instances over five independent runs. The test log-likelihood was estimated using 500 importance-weighted samples as done in Burda et al. (2016). The results are given in Table 1. Unsurprisingly, SIReN-VAE generally outperforms the vanilla VAE—this can be attributed to the more complex prior and posterior distributions introduced by the flows. However, we do not see any clear preference between the SIReN-VAE models. The picture changes when we consider the various models' performance when trained on much smaller training sets consisting of only $2 \times |\mathcal{G}|$ instances. Here, SIReN-VAE$_{\text{true}}$ clearly outperforms the other models. Figure 2 illustrates how SIReN-VAE$_{\text{true}}$ achieves a considerably lower loss than the other models for smaller training set sizes. We speculate that the increased sparsity of the neural network weights, in line with the true BN independencies, poses an easier learning task than those posed by the other models.

## 4 CONCLUSION

We propose the SIReN-VAE as an approach to incorporating an arbitrary BN dependency structure into a VAE. Including domain knowledge about the conditional independencies between observed and latent variables leads to much sparser weights matrices in both the encoder and decoder networks, but still allows sufficient information to propagate in order to model the data distribution. Indeed, our empirical results show that the sparsity induced by true conditional independencies is especially beneficial in settings where limited training data is available.

In this work we only considered synthetic datasets. Initial results on real-world datasets did not show similar performance gains when using a SIReN-VAE encoding the *hypothesized* BN structure. This could indicate that our approach is not sufficiently robust against deviations from the true underlying Bayesian network. Going forward, we hope to develop our understanding of the behaviour of SIReN-VAE when the encoded structure may be similar to, but not necessarily the same as, the data. This will be valuable for understanding how to use this approach effectively with real-world data, where the exact true BN structure is almost never available.

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

# A  SIReN-VAE Illustration

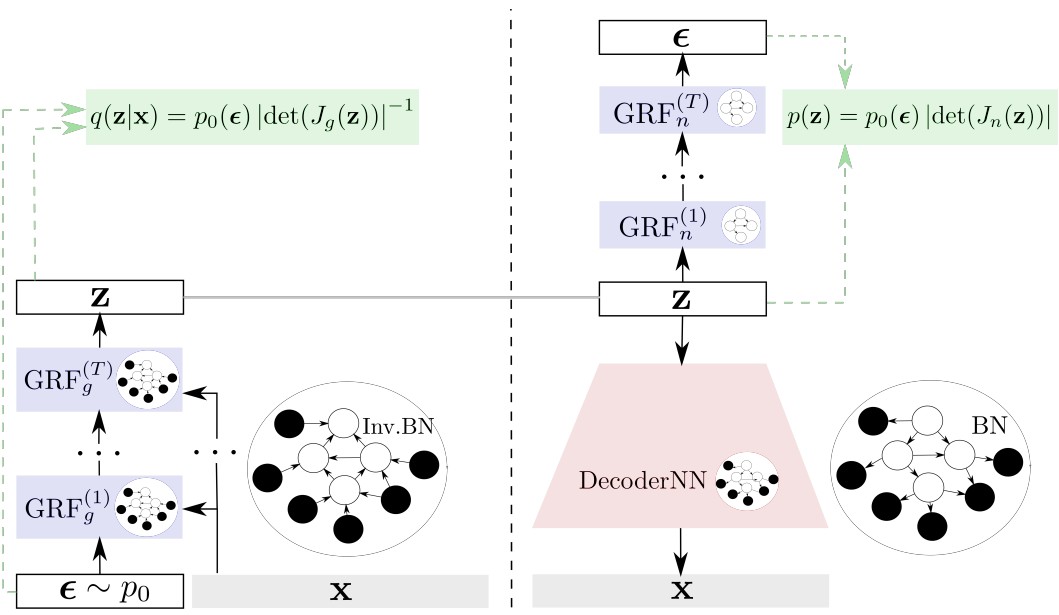

Figure A.1: SIReN-VAE encodes the BN's graphical structure into the decoder (right) via masking of the normalizing GRF ($\text{GRF}_n$) and decoder neural network weights. The inverted BN structure is similarly encoded in the inference network's generating GRF ($\text{GRF}_g$; left). The base distribution $p_0$ is chosen to be $\mathcal{N}(\mathbf{0}, I)$.

