# OpenReview forum: "SIReN-VAE: Leveraging Flows and Amortized Inference for Bayesian Networks"
_ICLR.cc/2022/Workshop/DGM4HSD — ICLR 2022 DGM4HSD workshop Poster_

### Official Review · Reviewer_AV4h · 2022-03-18
**Interesting way to add structure to VAEs**

**Rating:** 7
**Confidence:** 5

**Review:**

Black-box generative modeling and density estimation is a difficult task. One way it can be made easier is through the addition of structure into models. This structure can be represented with prior knowledge of conditional independence over the variables of the model. Latent-variable models have had difficulty in the past due to known issues such as posterior collapse. This work presents a way to incorporate prior knowledge of this independence structure by applying it explicitly through a normalzing flow (graph residual flow) which enforces group independence following a pre-specified bayesian-network structure. This structure is encoded into both the generator network and the inference network (where the structure of the true-posterior is used and derived from the bayesian network). The authors find that incorporating the structure of these models can lead to improved performance when the model's structure matches that of the data-generating process.

I would be very curious to see how this kind of structure can be used to deal with data where the ground truth network structure is unknown. For example, could we place a BN structure on a model which might encourage us to discern that an image contains distinct objects?

Overall, this was a well-written paper and I advocate for its acceptance to this workshop.

One nitpick is that you might want to consider changing the name as I thought I was about to read a paper about implicit function representations!!! https://arxiv.org/abs/2006.09661

---

### Official Review · Reviewer_Tou3 · 2022-03-25
**Incorporates known dependency structure into VAEs, lacking comparisons with alternatives and study of learned latent variables**

**Rating:** 6
**Confidence:** 3

**Review:**

# Summary
This work proposes a method for incorporating a known generative structure, in the form of the acyclic dependency graph of a Bayesian network (BN), into the variational autoencoder (VAE) framework. This allows for latent variable discovery given only the dependency structure of a BN without being given the exact data generating process. The BN dependency graph determines the structure of both the prior (in the form of a normalizing flow) and the decoder (a standard neural network), while the inverse BN graph determines the structure of the encoder (another normalizing flow). In particular, the authors choose to use a graphical residual flow, i.e. a residual flow with masking to incorporate the BN dependency structure, for both the prior and encoder. They demonstrate the effectiveness of this approach in comparison with a vanilla VAE on several datasets with known BN structures and achieve better results in the low data regime.

# Strengths, Weaknesses, & Questions
* This work provides a nice way of incorporating known dependency structures into VAEs. It makes sense that introducing additional inductive biases (in the form of a BN graph) should be effective, especially in the low data regime.
* Because the details of the architecture rely on graphical residual flows, which are also under review at this time, it is difficult to judge this work in isolation. It would be nice to see a comparison using some of the previously proposed graphical normalizing flows (e.g. Wehenkel or Weilbach) within the same VAE framework.
* As the authors note, the vanilla VAE may well be performing poorly simply because a lack of a complex prior/posterior. While variants of the proposed architecture are a nice test to demonstrate the usefulness of incorporating the BN, I would like to see a direct comparison against another VAE variant that uses normalizing flows (e.g. Kingma, 2016) and other VAEs that use structured priors & encoders like Ladder VAE (Sønderby, 2016) and Graph VAE (He & Gong, et al., ICLR 2019).
* Graph VAE (He & Gong, et al., ICLR 2019) is also able to use (and even discover) a dependency structure for the prior & encoder. Does using a normalizing flow improve on this significantly?
* Is the BN dependency structure necessary in all parts of the VAE? What happens if you only use BN for the prior & encoder?
* What would happen if you only had partial information or guesses as to what the dependency structure is (i.e. how well does it handle errors in the BN graph)? Would this approach also be able to discover the BN structure, or at least allow for a looser way of imposing the BN graph to allow for learned alterations to the connectivity or new unknown latent variables?
* In addition to performing well in terms of NLL and reconstruction error, I would like to see a more detailed study of the learned latent variables. Given that the hypothesis is that the BN provides an inductive bias for the dependency structure which helps with learning the right latent variables, it is would be very nice to see if the VAE has actually learned the expected latent variables in an interpretable way (especially in the low data regime). It would be great if you could train such a model on a new dataset and then interpret the learned latent variables afterwards.

# Conclusion
The authors have proposed a nice concept for enhancing a VAE with known dependency information and have shown a performance enhancement in the low data regime, but have not performed necessary comparisons/baselines nor demonstrated that their approach actually discovers useful and interpretable latent variables. Additional baselines are necessary to cement the demonstrated performance enhancement and justify the architecture choices.

---

### Decision · Program_Chairs · 2022-03-27

Accept (Poster)